# Reliability of the Coimbra Reactive Agility Soccer Test (CRAST)

**DOI:** 10.3390/jfmk8010011

**Published:** 2023-01-18

**Authors:** António Nóbrega, Hugo Sarmento, Vasco Vaz, Vítor Gouveia, Joel Barrera, Andreia Martins, Tomás Santos, João Pedro Duarte

**Affiliations:** University of Coimbra, Research Unit for Sport and Physical Activity (CIDAF), Faculty of Sport Sciences and Physical Education, 3040-256 Coimbra, Portugal

**Keywords:** technique, speed, nonplanned agility, field testing

## Abstract

Agility is a fitness-skill-related component that should be a part of the standard physiological testing for soccer players and one of the key performance indicators in soccer. The present study aimed to assess the reliability of the CRAST as a research tool in the study of soccer skills. Twenty-one university soccer players (chronological age: 19.3 ± 1.4 years; body mass: 69.6 ± 8.2 kg; stature: 173.5 ± 6.5 cm; federated training experience: 9.7 ± 3.6 years) volunteered for the testing protocol. The CRAST requires players to complete random courses six times as quickly as possible. In addition, the CRAST requires players to control and dribble the markers (four different colors: green, yellow, blue, and red). The soccer players completed three trials, each separated by one week. The first trial accounted for familiarization; the second and third were considered for analysis. The correlation for overall performance was very strong. The reliability of the CRAST was slightly better for total time than that for the penalty score (0.95 vs. 0.93). The TEM and the associated CV range of 7.04%–7.54% were for the penalty score and the total time, respectively. For both measurements, the ICC values also represent excellent reliability, as both values were over 0.900. The CRAST is a reliable protocol for assessing agility in soccer players.

## 1. Introduction

Talent identification programs play an essential role in sports, aiming to identify gifted players for high-level practice through clubs and academies [1]. Soccer requires a multivariate approach for the early detection of talented players [2,3]. Previous proposals included anthropometry and body composition measurements [3,4], physical and fitness performance [2,4], psychological tests [5], and skill tests [2,3,4,6].

Coordinative capacity agility is one of the key performance indicators in soccer. It is a fitness-skill-related component that should be a part of standard physiological testing for soccer players [7]. Soccer match performance indicates that the game is characterized by fast movements that become prominent in short and long sprints, explosive reactions (such as jumping), and quick changes in direction [8]. Short-term maximal actions impact soccer performance by requiring speed, acceleration, or agility [9]. Agility is defined as rapid whole-body movement with a change in speed or direction in response to a stimulus [10]. Agility has relationships with trainable physical qualities such as strength, power, and technique, and cognitive components such as eye-tracking techniques, visual scanning speed, and anticipation [10].

Agility testing is challenging to quantify since it is a physical component that is influenced by many factors (internal and external), and there is difficulty in providing a test with enough uncertainty to replicate game conditions and enough precision in the protocol for the test to be reliable [1]. Considering this, many tests used to measure agility evaluate other physical capacities in team sports, such as the capacity for a previously planned change in direction and coordination. From this perspective, the literature has numerous evaluation tests, such as the 5–0–5 agility [11], pro agility [12], Illinois agility [13], arrowhead agility [14], and ladder agility [15] tests. Furthermore, static rather than dynamic drills or opposition can blur the distinction between ‘‘technique’’ and ‘‘skill’’. The specific feature of a skilled movement is that the player has a learned ability to select and with which to perform the correct technique as determined by the demands of the situation. Recognizing the limitations of the available tests, we developed the Coimbra Reactive Agility Soccer Test (CRAST) to assess the multifaceted aspects of soccer skills, including dribbling, control, and decision making. Thus, the present study aimed to assess the reliability of the CRAST as a research tool in the study of soccer skills.

## 2. Materials and Methods

### 2.1. Participants

In total, 21 male university soccer players (age: 19.3 ± 1.4 years; stature: 173.5 ± 6.5 cm; body mass: 69.6 ± 8.2 kg) took part in this prospective study. Chronological age was determined to the nearest 0.1 years by subtracting the birth date from the date of the first testing measurement. Training experience was recorded via questionnaire and confirmed in Portuguese federation records. In a two-game week (Sunday to Sunday matches), players participated in 4–5 h/week of formal soccer training (three training sessions). All participants were informed of the purpose and content of the study, and provided written informed consent before participation.

### 2.2. Procedures

The study followed the ethical standards for sports medicine with human samples [16]. The study was approved by the Scientific Council and the ethics committee of the University of Coimbra (CE/FCDEF-UC/00692021), and conducted according to the Declaration of Helsinki. Tests were performed on the same weekday and at the same period (i.e., 16:00–18:00) to avoid variation due to the circadian rhythm. Test conditions were controlled, and test sessions were performed on artificial grass under similar conditions following the same warming-up routine (a 10 min standardized warm-up consisting of jogging, striding, sprinting, and stretching exercises preceded the trials prescribed by the same observer), always under the guidance of the same observers. The soccer players completed three trials, each separated by one week. The first trial accounted for familiarization; the second and third were considered for analysis.

### 2.3. Anthropometry and Body Composition

Stature was measured to the nearest 0.1 cm using a Harpenden stadiometer (model 98.603, Holtain Ltd., Crosswell, UK). Body mass was recorded using a scale SECA (model 770, Hanover, MD, USA) with a 0.1 kg reduction. The thickness of seven skinfolds (biceps, triceps, subscapular, abdominal, suprailiac, thigh, and calf) was measured following the recommendations from the International Society for the Advancement of Kinanthropometry (ISAK) [17] using a Lange caliper (Beta Technology, Ann Arbor, MI, USA). Body fat percentage was calculated from the measurements of the standard skinfold equation for the antithetic population [18].

### 2.4. Yo-Yo Intermittent Recovery (IR) Test Level 1

Yo-Yo IR1 was conducted according to the original research guidelines [19]. Participants were instructed to refrain from strenuous exercise for at least 48 h and to consume their regular pretraining diet before the test sessions. A standardized warm-up preceded each Yo-Yo IR1. All tests were completed in an indoor pavilion with a temperature of around 20 °C. All players ran the test with running shoes. The total duration of the test was around 25 min, and the individual scores are expressed as the covered distance (m). VO_2max_ was derived from the following formula [20]:VO_2max_ (mL/min/kg) = IR1 distance (m) × 0.0084 + 36.4(1)

### 2.5. Coimbra Reactive Agility Soccer Test (CRAST)

Soccer players completed three trials, each separated by one week. The first trial accounted for familiarization; the second and the third trial were considered for analysis. Figure 1 illustrates the layout of the CRAST. Colored cones were used to distinguish the different zones, with a white cone in the middle of the CRAST area representing the start and finish of the test. The test had four journeys identified with four different colors. There were two distances, five and seven meters, and it was required that the soccer players completed random courses (an observer identified the randomly preselected colors) six times, as quickly as possible. The participants began with a ball (model Madrid 2020, size 5, Mka Ltd., Famalicão, Portugal) in the central white cone, and the first examiner started timing the test using a hand-held stopwatch (model AX602 Dual 100, Accusplit Ltd., Pleasanton, CA, USA). The second examiner called out the color list, with this being the next direction called just before the participant completed the current course. The same examiner was used in each role to eliminate interexperimenter variability. The order of the journeys was randomly generated for each trial. Each trial session consisted of three long (green and yellow) and three short (blue and red) courses. The participants were informed that ball control could be executed with both feet in the testing area between the marked lines (see Figure 1). The second examiner stopped the clock when the player had returned to the white cone, and was also responsible for recording penalty time points accrued during the trials. Thus, the examiner stood in such a position that all four target areas could be viewed. Penalty time was awarded according to previous research [6] and the following errors: five second for taking the wrong color course, and one second for controlling the ball from outside of the designated area.

### 2.6. Statistics

Descriptive statistics are reported: range, mean value (standard error and the respective 95% confidence interval (CI)), and standard deviation as a measure of dispersion. Normality was checked with the Shapiro–Wilk test, and when the premises had been violated, a logarithmic transformation was performed to reduce the nonstandardized error. An initial bivariate correlation was established between the two trials. The reliability and intraclass correlation coefficient (ICC) were calculated in parallel to the technical error of measurement (TEM) [21]. The coefficient of variation (% CV and respective 95% confidence intervals) was expressed as the percentage of the mean. All statistical analyses were performed using IBM SPSS v.26 for Mac OS software (SPSS Inc., IBM Company, Armonk, NY, USA). Subsequently, data were inspected using Bland-Altman plots that combined the errors against the mean derived from two different moments using Graphpad Prism software (GraphPad Software, Inc.; La Jolla, CA, USA).

## 3. Results

Descriptive statistics are presented in Table 1, including decimal age, formal training experience, anthropometry, body composition, aerobic performance derived from the Yo-Yo protocol, and performance scores resulting from the CRAST protocol. Except for the chronological age and percentage of fat mass, all variables fit the normal distribution.

A summary of the CRAST performance scores is presented in Table 2. The performance score is the junction of two variables, the time taken to complete the CRAST (total time), and any accrued penalty time for poor control (penalty of 1 s) or wrong course round (penalty of 5 s). Trial 2 and 3 scores were slightly improved in the total time compared to those of Trial 1. However, no improvement was detected for the mean of penalties.

Table 3 reports that the reliability of the CRAST was slightly better for total time than that for the penalty score (0.95 vs. 0.93). The correlation for the overall performance was very strong. Furthermore, although calculated differently, the intraclass correlation coefficients for the data show nearly identical results as Pearson’s correlation. TEM and the associated CV range between 7.04–7.54% were for the penalty score and the total time, respectively. For both measurements, the ICC values also represent excellent reliability, with both values being over 0.900.

Figure 2 presents the agreement of repeated measures for the total time and the penalty score was computed through the Bland-Altman analysis for the total time (Bias = 0.004; lower limits of agreement [LLOA] = −1.622; and upper limits of agreement [ULOA] = 1.630) and for the penalty score (Bias = −0.140; LLOA = −2.042; ULOA = 1.761).

## 4. Discussion

The aim of the present study was to test the reliability of the CRAST parameters, as performed by experienced soccer players as tools to assess soccer skill for research training and research purposes. The main finding was that the CRAST was highly reliable on both variables, evidencing that this methodology is a good tool for monitoring agility during training or assessment routines. Although talent identification processes are used within elite academies to discover future players, the process can also be undertaken at lower levels, such as regional school centers and local clubs that may need to evaluate or grade players [1]. Regardless of the population tests and the tested ability, one maximal test trial should precede the testing, reducing particular motor learning effects [22]. Then, simply reporting the correlation may be an insufficient reliability indicator [23]. Thus, the test–retest correlations in the present study, r = 0.95 (for the total time) and r = 0.93 (for the penalty score), were consistent with previous studies [22], considering 150 youth soccer players. Within-subject variations (CV) are also acceptable (7.3%), this being in line with previous research (i.e., 5.6% [22] and 8.0% [24]). Our results suggest that the CRAST could be used for this purpose because it is highly reliable between trials with amateur soccer players with deliberate and federated practice.

Many related physical tests intend to assess the previously determined ability to change direction, not the agility skill. However, more recent tests are starting to have a better conceptual apparatus, which includes the agility component: the reactive agility test [10], the butterfly agility test [25], and the football specific reactive agility test [26]. However, most of these tests only examine the ability to quickly change direction (i.e., COD) without including the response to unpredictable external stimuli, which is an essential facet of agility [26]. Therefore, there is a clear distinction between changing direction, which is preplanned, and reactive and nonplanned agility, especially in detecting cognitive (i.e., perceptual, decision making) and physical (i.e., conditioning capacities) determinants of agility [27]. So far, perceptual and cognitive capacities have been identified as significant predictors of reactive agility in team sport athletes [27,28].

Agility is crucial for long-term soccer development, and studies frequently focused on the reliability and validity of the tests aiming to test, retest, and compare this capacity among different ages and competitive levels of expertise [29,30]. For example, a previous study [31] examined the reliability and validity of the modified Barrow COD test, and reported similar high test–retest reliability (ICC: 0.94) for 11-to-14-year-old soccer players. A recent study [8] considered 20 U17 and U19 male soccer players divided into three playing positions (defenders, midfielders, and forwards), and presented appropriate-to-high (ICC: 0.70 to 0.92) reliability in reactive agility and COD testing, respectively. Lastly, considering adult athletes, performing the butterfly agility test (BAT) [25] had a good reliability value (ICC = 0.89). Agility is essential in soccer [32], involving dribbling, passing, and kicking, and indicating the importance of ball control [32]. Consequently, tests designed to simulate real-game situations in soccer are increasing in the literature [33,34]. Thus, researchers have developed soccer-specific COD drills and reactive agility tests, including ball control [8,35].

Previous studies indicated that sport-specific agility tests should consider the specificity of the movement technique [8]. For example, soccer players must repeatedly change direction with various jumps, breaks, accelerations, and decelerations. Thus, they often perform turns, alternate between running and lateral shuffling, and change from forward to backward running (frontiers). Previous authors [36] suggested that the Y-shaped course may not be appropriate reactive agility to evaluate soccer players. So, it becomes evident that COD and reactive agility should be considered to be vital components for successful performance in soccer [8]. More recently, a study [8] investigated soccer-specific reactive agility where players performed agility tests by adding the soccer-specific movement of kicking. In brief, tests showed high reliability and power to discriminate between playing levels (e.g., U17 vs. U19 players) [8]. However, due to the absence of soccer-specific reactive agility tests that involve specific stop-and-go movement patterns and ball techniques, the primary rationale for this study was to determine whether newly developed tests of reactive agility would be reliable in evaluating soccer-specific agility performance.

The main limitation of this study originates from the cross-sectional design. The current study only observed one performance-level group involved in equal sports settings. Furthermore, the chronological age range reflects other noncontrolled factors (i.e., the initial selection of players in different generations). Moreover, the primary purpose was to develop and evaluate a soccer-specific test of reactive agility, including the dribbling ability often executed in real-game contexts. Therefore, although the presented and evaluated tests exhibited very good reliability, further studies are required to investigate dribbling-specific soccer agile performance at different competitive levels.

## 5. Conclusions

In summary, the results of the present study demonstrated that the CRAST is an easy-to-apply and low-cost methodology with high reliability. This tool is a sport-specific field test of reactive agility in amateur soccer players, with the outcomes, total time, and penalty score able to monitor reactive agility in soccer. Moreover, the results should be of interest to coaches, sports scientists, and others involved in the selection and development of soccer. Future studies should consider younger ages and compare different levels of expertise.

## Figures and Tables

**Figure 1 jfmk-08-00011-f001:**
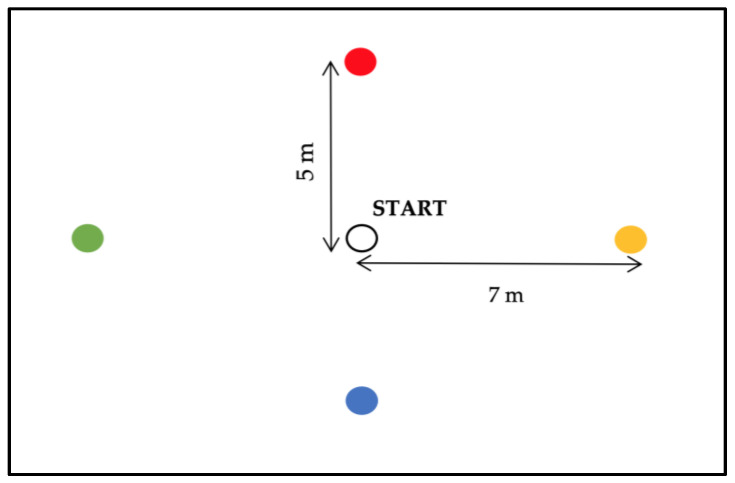
Schematic representation of the Coimbra Reactive Agility Soccer Test (CRAST). Colored cones were used to distinguish the different zones (7 m [long]: yellow and green; 5 m [short]: blue and red), with a white cone in the middle representing the start and finish of the test.

**Figure 2 jfmk-08-00011-f002:**
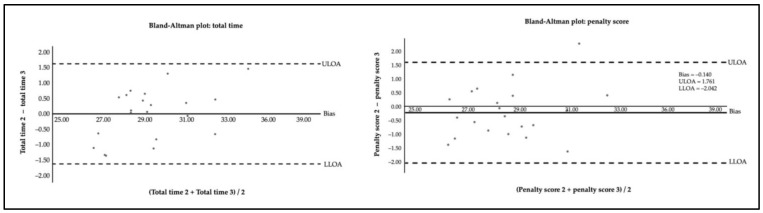
Agreement of repeated measures for the total time and the penalty score. The dashed lines represent 95% limits of agreement (±1.96 SD), lower limits of agreement (LLOA), and upper limits of agreement (ULOA).

**Table 1 jfmk-08-00011-t001:** Descriptive statistics for the total sample (n = 21).

Variable	Range		Mean			SD	Shapiro–Wilk	
	Minimum	Maximum	Value	SEM	(95% CI)		Value	*p*
Chronological age (years)	18.2	24.6	19.3	0.3	(18.8; 20.0)	1.4	0.637	<0.001
Training experience, (years)	2.0	15.0	9.7	0.8	(8.1; 11.1)	3.6	0.942	0.235
Stature (cm)	159.0	186.5	173.5	1.4	(170.8; 176.1)	6.5	0.972	0.779
Body mass (kg)	55.6	86.5	69.6	1.8	(66.2; 73.0)	8.2	0.960	0.524
Fat mass (%)	9.0	21.0	12.7	0.7	(11.5; 14.2)	3.1	0.839	0.003
Yo-Yo IR1 (m)	320	1100	696	199	(605; 784)	43	0.977	0.871
Yo-Yo IR1, VO_2max_ (mL/kg/min)	39.1	45.6	42.2	0.4	(41.5; 43.0)	1.7	0.977	0.868
Trial 1 CRAST total time (s)	27.86	35.22	30.47	0.47	(29.63; 31.37)	2.14	0.912	0.060
Trial 2 CRAST total time (s)	26.76	36.69	30.47	0.55	(29.47; 31.52)	2.52	0.942	0.235
Trial 1 CRAST penalties (#)	0	2	0.6	0.1	(0.3; 0.9)	0.7	0.950	0.348
Trial 2 CRAST penalties (#)	0	2	0.8	0.1	(0.3; 0.9)	0.8	0.944	0.263

Abbreviations: IR, intermittent recovery; CRAST, Coimbra Reactive Agility Soccer Test; SEM, standard error of the mean; CI, confidence intervals; SD, standard deviation; #, number.

**Table 2 jfmk-08-00011-t002:** Total time and penalties mean values for the three trials.

Variable	Familiarization Mean	Trial 2 Mean	Trial 3 Mean
Total time (s)	30.60	30.47	30.47
Penalties (#)	1	1	1

#, number.

**Table 3 jfmk-08-00011-t003:** Reliability, technical error of measurement, coefficient of variation, and intraclass correlation between Trials 2 and 3.

Variable	r	TEM	CV (%)	ICC
Total time (s)	0.95 **	0.54	7.54	0.967
Penalty score (#)	0.93 **	0.54	7.04	0.944

Abbreviations: R, reliability; TEM, technical error of measurement; CV, coefficient of variation; ICC, intraclass correlation. ** Significant correlation between trials. #, number.

## Data Availability

Not applicable.

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
