# Peer review of "Reliability of the Coimbra Reactive Agility Soccer Test (CRAST)"

_jfmk, 2023, doi:10.3390/jfmk8010011_

Round 1

Reviewer 1 Report

The manuscript is quite well designed and the study methods perfectly executed. There are only a couple of suggestions to imrpove it

1) Include limititations in the final section of the discussion

2) In table 2 indicate the units between brackets

3) Minor corrections in English expressions

- Line 136: improve the sentence for a better understanding

- Line 192: improve the sentence for a better understanding

Author Response

REVIEWER 1

The manuscript is quite well designed and the study methods perfectly executed. There are only a couple of suggestions to improve it.

Authors: Your comments are acknowledged.

1) Include limitations in the final section of the discussion.

Authors:

Thank you for the remark. The last paragraph of the discussion includes the limitations.

2) In table 2 indicate the units between brackets

Authors: Thank you. It was adjusted tin the three tables.

3) Minor corrections in English expressions

- Line 136: improve the sentence for a better understanding

Authors: The sentence was rephrased.

- Line 192: improve the sentence for a better understanding

Authors: The sentence was rephrased.

Reviewer 2 Report

This study aims to examine the reliability of the Coimbra Agility Soccer Test. The idea is good. However, the study is poorly executed, with some serious methodological flaws. My comments are presented in the additional text.

Title:

·       Since you tried to evaluate the Reactive agility test, this should be in the name of the test.

Introduction:

·       The study rationale is not entirely clear. Since we have so many different agility tests for football, you need to provide sufficient elaboration (in one additional paragraph) where are the flaws of the commonly used tests. Otherwise, there is no needs to develop yet another agility test.

·       Furthermore, since you tried to develop a reactive agility test, the introduction needs more elaboration on this topic (i.e., differences between agility and reactive agility).

Methodology:

·       Bland-Altman plots should also be used in reliability studies.

·       This study lacks validity testing (often done with reliability). What is the point of having a reliable test if it is not valid?

·       Why did you perform the Yo-Yo test? There are no indices in the Introduction that this test will be used. Also, in the Methodology section, nothing indicates the need for this test. It just appears out of nowhere.

·       I believe that this test should be evaluated without the ball first. That way, the application of this test might extend to other sports, thus increasing the use of the test.

·       The are three major flaws with this test:

o   Testing should be performed on natural grass to increase the ecological validity of the test.

o   The use of a stopwatch is questionable. Although the test proved reliable, the concurrent validity of the stopwatch vs. time gates (for this test) was necessary to precede this study.

o   The most critical problem with this test is: “The second examiner was involved in calling out the colors list being the next direction called just before the participant completed the current course.”. This must be automatically done with some technical mechanism. There is a big chance for error when somebody subjectively gives commands.

Discussion and conclusions:

·       There are numerous methodological flaws in this study. Therefore, I won’t comment on the discussion and the conclusion.

Author Response

Title:

Since you tried to evaluate the Reactive agility test, this should be in the name of the test.

Thank you. Altered.

Introduction:

The study rationale is not entirely clear. Since we have so many different agility tests for football, you need to provide sufficient elaboration (in one additional paragraph) where are the flaws of the commonly used tests. Otherwise, there is no needs to develop yet another agility test.

Furthermore, since you tried to develop a reactive agility test, the introduction needs more elaboration on this topic (i.e., differences between agility and reactive agility)

Thank you for the pertinent comment. The introduction includes the rationale of the study. Also, the discussion section includes the difference between agility and reactive agility (Line 190).

Methodology:

Bland-Altman plots should also be used in reliability studies.

Authors: Thank you for the pertinent insight. Included.

This study lacks validity testing (often done with reliability). What is the point of having a reliable test if it is not valid?

Authors: Appreciated. This is a limitation of the study (we assume that in the last paragraph of the discussion section). As described in the methods section, this sample belongs to the University championship, so, unfortunately, authors do not have access to different competitive levels to proceed with the validity of the test. Although we intend to study the validity of the test in future research, also associate mature status with youth soccer players.

Why did you perform the Yo-Yo test? There are no indices in the Introduction that this test will be used. Also, in the Methodology section, nothing indicates the need for this test. It just appears out of nowhere.

Authors: The Yo-Yo test was introduced as other variables (i.e., body composition) to characterize the sample.

I believe that this test should be evaluated without the ball first. That way, the application of this test might extend to other sports, thus increasing the use of the test.

Authors: We agree with your opinion.

There three major flaws with this test:

Testing should be performed on natural grass to increase the ecological validity of the test.

Authors: Thank you for the observation. This group of University soccer players is used to train and compete on artificial turf.

The use of a stopwatch is questionable. Although the test proved reliable, the concurrent validity of the stopwatch vs. time gates (for this test) was necessary to precede this study.

Authors: Thank you for the remark. A similar stopwatch was used in previous reference published article:

Ajmol Ali , Clyde Williams , Mark Hulse , Anthony Strudwick , Jonathan Reddin , Lee Howarth , John Eldred , Matthew Hirst & Steve McGregor (2007) Reliability and validity of two tests of soccer skill, Journal of Sports Sciences, 25:13, 1461-1470, DOI: 10.1080/02640410601150470

The most critical problem with this test is: “The second examiner was involved in calling out the colors list being the next direction called just before the participant completed the current course.”. This must be automatically done with some technical mechanism. There is a big chance for error when somebody subjectively gives commands.

Authors: Appreciated. Also, as answered previously, similar procedures were assumed in previous reference published article:

Ajmol Ali , Clyde Williams , Mark Hulse , Anthony Strudwick , Jonathan Reddin , Lee Howarth , John Eldred , Matthew Hirst & Steve McGregor (2007) Reliability and validity of two tests of soccer skill, Journal of Sports Sciences, 25:13, 1461-1470, DOI: 10.1080/02640410601150470

Discussion and conclusions:

There are numerous methodological flaws in this study. Therefore, I won’t comment on the discussion and the conclusion.

Authors:

Reviewer 3 Report

Basic reporting

The study reported the reliability of the Coimbra Agility Soccer Test as a research tool in the study of soccer skills. Whilst the study undoubtedly has merit, there are some aspects that need clarification, to improve the readability of the manuscript.

ABSTRACT

Please, introduce the article's purpose in the abstract.

MATERIAL AND METHODS

Participants

Please introduce the sample’s training volume per week (hours x week).

Moreover, I suggest you improve the description of the sample selection. Please, could you deeply present how you proceed to select your sample? 

It is a representative sample? If is it, introduce please the Sample size calculation.

Procedures

Please present information about how and where tests were performed (kind of warm-up performed, who organized and applied the test, etc.). Please detail all the main information and the necessary details in order to provide the reader with a clear picture of how tests were performed

L-211-218: I suggest you to included more considerations about the limitation of your study.  Please add which are the possible future lines.

CONCLUSIONS

The conclusion section should be further developed. Moreover, a practical implications section should be included. How these findings can help soccer coaches and practitioners? Which practical implications can it have in training sessions? 

Author Response

ABSTRACT

Please, introduce the article's purpose in the abstract.

Authors: Included.

MATERIAL AND METHODS

Participants

Please introduce the sample’s training volume per week (hours x week). Moreover, I suggest you improve the description of the sample selection. Please, could you deeply present how you proceed to select your sample? It is a representative sample? If is it, introduce please the Sample size calculation.

Authors: Thank you for the pertinent comments. The training volume was included. The sample was selected by convenience. These players belong to the university soccer team.

Procedures

Please present information about how and where tests were performed (kind of warm-up performed, who organized and applied the test, etc.). Please detail all the main information and the necessary details in order to provide the reader with a clear picture of how tests were performed

Authors: Thank you for the remark. Detailed information was included in the procedures.

L-211-218: I suggest you to included more considerations about the limitation of your study. Please add which are the possible future lines.

Added.

CONCLUSIONS

The conclusion section should be further developed. Moreover, a practical implications section should be included. How these findings can help soccer coaches and practitioners? Which practical implications can it have in training sessions?

Thank you. Practical applications were included.

Round 2

Reviewer 2 Report

/